# Applicability and Validity of Second Victim Assessment Instruments among General Practitioners and Healthcare Assistants (SEVID-IX Study)

**DOI:** 10.3390/healthcare12030351

**Published:** 2024-01-30

**Authors:** Stefan Bushuven, Milena Trifunovic-Koenig, Maxie Bunz, Patrick Weinmann-Linne, Victoria Klemm, Reinhard Strametz, Beate Sigrid Müller

**Affiliations:** 1Training Center for Emergency Medicine (NOTIS e.V), 78234 Engen, Germany; milena.trifunovic-koenig@notis-ev.de; 2Department of Anesthesiology and Critical Care, Faculty of Medicine, Medical Center—University of Freiburg, University of Freiburg, 79106 Freiburg, Germany; 3Institute for Medical Education, University Hospital, LMU Munich, 80539 Munich, Germany; 4Wiesbaden Institute for Healthcare Economics and Patient Safety (WiHelP), Wiesbaden Business School, RheinMain University of Applied Sciences, 65183 Wiesbaden, Germany; victoria.klemm@hs-rm.de (V.K.); reinhard.strametz@hs-rm.de (R.S.); 5Institute of General Practice, University of Cologne, 50937 Cologne, Germany; maxie.bunz@uk-koeln.de (M.B.); patrick.weinmann-linne@uk-koeln.de (P.W.-L.); beate.mueller@uk-koeln.de (B.S.M.)

**Keywords:** psychological trauma, moral distress, second victim, resilience, team factors, human resource, moral injury, patient safety, healthcare worker safety, general practitioners, GP

## Abstract

Background: The second victim phenomenon and moral injury are acknowledged entities of psychological harm for healthcare providers. Both pose risks to patients, healthcare workers, and medical institutions, leading to further adverse events, economic burden, and dysfunctionality. Preceding studies in Germany and Austria showed a prevalence of second victim phenomena exceeding 53 percent among physicians, nurses, emergency physicians, and pediatricians. Using two German instruments for assessing moral injury and second victim phenomena, this study aimed to evaluate their feasibility for general practitioners and healthcare assistants. Methods: We conducted a nationwide anonymous online survey in Germany among general practitioners and healthcare assistants utilizing the SeViD (Second Victims in Deutschland) questionnaire, the German version of the Second Victim Experience and Support Tool Revised Version (G-SVESTR), and the German version of the Moral Injury Symptom and Support Scale for Health Professionals (G-MISS-HP). Results: Out of 108 participants, 67 completed the survey. In G-SVESTR, the collegial support items exhibited lower internal consistency than in prior studies, while all other scales showed good-quality properties. Personality traits, especially neuroticism, negatively correlated to age, seem to play a significant role in symptom count and warrant further evaluation. Multiple linear regression indicated that neuroticism, agreeableness, G-SVESTR, and G-MISS-HP were significant predictors of symptom count. Furthermore, moral injury partially mediated the relationship between second victim experience and symptom count. Discussion: The results demonstrate the feasible use of the questionnaires, except for collegial support. With respect to selection bias and the cross-sectional design of the study, moral injury may be subsequent to the second victim phenomenon, strongly influencing symptom count in retrospect. This aspect should be thoroughly evaluated in future studies.

## 1. Introduction

### 1.1. Background and Rationale

Working in complex medical environments exposes healthcare providers (HCPs) to the risk of emotional traumatization. Predominantly, though not exclusively, following medical errors, the experience of guilt and shame in responsible or extreme situations may trigger the onset of a second victim phenomenon (SVP) [1,2,3,4,5]. The European Researchers‘ Network Working on Second Victims (SVs) defines this phenomenon as “*Any healthcare worker, directly or indirectly involved in an unanticipated adverse patient event, unintentional healthcare error, or patient injury, and who becomes victimized in the sense that they are also negatively impacted*” [4]. SVP is known to cause severe reactions in HCPs and may lead to absenteeism, post-traumatic stress disorder (PTSD), depression, and even suicide [6]. On an institutional level, SVs who cannot cope with the situation may leave their institutions, the medical system as a whole, or work in a dysfunctional manner [7], causing further harm to patients and institutions, e.g., by practicing defensive medicine [8,9,10,11]. Consequently, the role of institutional support in prevention and intervention, especially in developing and maintaining safety culture, is crucial [12].

Recent work on SVP in Germany and Austria revealed high prevalences for the reported lifetime occurrence, affecting 59 percent of German physicians [13], 60 percent of German nurses [14], 53 percent of German emergency physicians [15], 89 percent of Austrian pediatricians [16], and 43 percent of HCPs in an Austrian hospital [17].

Moral injury (MI) is a phenomenon defined as an acute violation of one’s ethical code. MI has primarily been described among military veterans [18], but there is growing evidence of its impact on healthcare workers [19,20]. MI, characterized by the acute transgression of moral beliefs, is an interconnected, subsequent, and concurrently developing entity to “moral dilemma” (two incongruent conflicting moral beliefs) [21] and “moral distress.” Moral distress refers to the conflict between current actions and a previously established moral decision or commitment. It develops due to either the unavailability of an option (constraint distress) or the unawareness of available options (uncertainty distress) [22,23]. The experience of MI, developing alongside or subsequent to moral distress, may lead to pathologies such as PTSD [21]. The clear interaction of SV, MI, and other entities within the moral complex (moral dilemma, moral distress, moral resilience, compassion fatigue, etc.) is yet to be fully understood.

Neuroticism was identified as a risk factor for developing SVP by the Second Victims in Deutschland (SeViD) studies [14,15,16]. These studies are primarily focused on the inpatient setting or emergency medicine. However, SVP and MI in the outpatient setting, particularly among general practitioners (GPs) and healthcare assistants (HCAs), have not yet been specifically examined. GPs constitute one of Germany’s largest groups of physicians frequently facing psychological distress [24,25], especially during the pandemic [26,27]. Nevertheless, the psychological burden they experience is underinvestigated [28] despite its high prevalence [29]. Hence, the exact role of SVP in the psychological burden on GPs remains unknown. Additionally, since the instruments for the detection and assessment of SVP [30,31,32] were originally developed and validated in hospital and emergency medicine environments, their usability for medical staff outside hospitals is unknown.

In prior studies conducted by our working group, we translated and validated the German versions of the Second Victim Experience and Support Tool Revised Version (G-SVESTR) and the Moral Injury Symptom and Support Scale for Health Professionals (G-MISS-HP) [31,32]. The G-SVESTR demonstrated good face, content, and factor validity, with reliability exceeding the benchmark at over 0.8 (Cronbach’s alpha (α) = 0.84) in the validation study [31]. The instrument consists of 35 items focusing on physical and psychological distress, support from colleagues, supervisors, and institutions, professional self-efficacy, turnover intentions, absenteeism, and resilience. In addition, the G-MISS-HP has demonstrated validity, including face, content, factor, and convergent validity, and reliability in a validation study (α = 0.79) [32]. The G-MISS-HP is available in both secular and nonsecular versions, consisting of 8 items in the secular and 10 items in the nonsecular version. Both versions encompass 8 items related to betrayal, guilt, shame, trouble, trust, meaning, forgiveness, and failure. Additionally, the nonsecular version includes two items addressing punishment and faith.

### 1.2. Objectives

This study aims to investigate the usability of the G-SVESTR [31], the G-MISS-HP [32], and the modified SeViD [30] questionnaires among GPs and HCAs in preparation for an extensive study involving one of the largest groups of German physicians with more than 120,000 members. Our objective is to ascertain the transferability of these instruments and, if necessary, identify essential modifications to the surveys related to structural properties or content validity. To test the usability and transferability of the instruments, we implemented an open-field entry method, providing participants with the opportunity to share their perspectives on individual items and report any challenges they encountered. We thoroughly analyzed participants’ feedback, identifying common themes, patterns, and specific concerns to gain insights into the usability of the G-SVESTR, G-MISS-HP, and SeViD from the users’ standpoint. In addition, we examined the responses provided by the participants to the items on the instruments. Our examination extended to exploring inter-item correlations, with a particular focus on identifying any item that exhibited distinctive characteristics. Certain aspects of the instrument may pose challenges for GPs and HCAs and confuse them in offering conclusive responses, as these items are more relevant to clinical settings, whereas GPs predominantly work in outpatient environments. Examining the transferability and usability of these instruments is essential to ensure their effectiveness in the outpatient setting of GPs and HCAs, and to minimize biases and errors by reducing the risk of participant confusion and contributing to the validity of research outcomes. Additionally, optimizing resource utilization is a key benefit, as usable instruments are easier and faster to use, increasing overall efficiency. Efficient resource use is particularly important in healthcare settings, where time and resources are often limited. Furthermore, our objectives encompassed determining correlations between demographic characteristics, intensity of experienced MI and SVP, and symptom count. This aimed to pave the way for developing new hypotheses concerning the SVP and MI and their manifestations in disparate situations, professions, and occupational settings—potentially prompting the need for different prevention and education frameworks.

In addition, previous SeViD studies have focused on identifying antecedents of symptoms that may occur as distinct consequences following an SV experience. Factors such as age, length of professional experience [15,16], personality traits (especially neuroticism, as previously noted) [14,15,16], and certain work-related characteristics, such as outpatient-sector involvement in Austrian pediatricians [16], have shown significance in these studies. However, until now, no one has explored the impact of MI on symptom count. Exploring the associations between MI and symptom count is important for gaining insights into the potential psychological and psychosomatic consequences and manifestations of MI. It allows for a better understanding of how experiences related to MI may contribute to or influence the development and severity of symptoms after an SV experience besides demographics and personality traits, providing valuable information for therapeutic interventions, support, and preventive measures. Hence, we aimed to test the impact of age, personality traits, SV, and MI on symptom count. Moreover, our objective encompassed determining the potential protective role of collegial, supervisory, and institutional support in the associations between G-SVESTR psychological and physical distress subscales and symptom count. This investigation may shed light on factors that can help in reducing and preventing further symptoms, which can be very adverse and can influence the wellbeing of healthcare professionals facing psychological or physical distress as a consequence of an SV experience. This understanding may have practical implications for intervention strategies.

## 2. Methods

### 2.1. Study Design, Setting, and Participants

We conducted a cross-sectional, anonymous, open, online survey between April and October 2023. Participants were recruited through various mailing lists from the investigators’ professional network, e.g., mailing lists of teaching and research practices. In addition, practice teams were invited to participate during regional and national conferences and training events for GPs in Germany. In total, the invitation was distributed to around 800 GP practices. We want to emphasize that, during the distribution of the invitation link, we did not possess specific demographic information about the GPs or HCAs who received it. Eligible participants included those working as GPs or HCAs, with GP trainees also qualifying for participation.

### 2.2. Variables

We used variables from surveys previously tested for validity and reliability in other healthcare settings [19,33,34,35,36], validated in the German language [30,31,32]. Modifications were made to the SeViD questionnaire to reduce response burden. The survey included:eight demographic items (gender, objective age, subjective age, occupational group, workplace, nation),an item about prior knowledge of the term “second victim” (yes/no),an item regarding previously completed specialization in palliative care (yes/no),the short German version of the Big Five Inventory (BFI-10) [37], assessing participants’ personality traits (openness, neuroticism, agreeableness, extraversion, and conscientiousness) on a 5-point Likert scale [37].The modified SeViD-questionnaire covered:
i.symptoms after a potentially traumatic situation with multiple-answer options (yes/no) for various possible symptoms such as anxiety about being excluded from work by colleagues, anxiety about getting fired, listlessness, depressive symptoms, lack of concentration, flashbacks during work, flashbacks outside of work, aggressiveness, defensive behavior, psychosomatic symptoms such as headache and back pain, sleeplessness, drug or alcohol abuse, feelings of shame, feelings of guilt, loss of self-esteem, social isolation, anger towards others, anger towards oneself, desire for support, and desire to better understand the incident,ii.duration of symptoms for more than 12 months (yes/no), andiii.complete recovery from symptoms (yes/no).iv.Symptom count was measured by summing the reported symptoms mentioned earlier.
Four free-text entries gathered information on the situation, symptoms, support received, interaction with the pandemic, and coping strategies in regards to SVP.Experience, symptoms, and support strategies of SVP were measured using the G-SVESTR comprising nine dimensions with three to five items each. We computed an overall G-SVESTR score as the sum of the nine partially recorded mean values of the subscales, deviating from previous studies by including this overall score. Notably, the support subscales of the G-SVESTR instruments were not included in the overall score [31].Symptoms and support options of MI were measured using the German version of the eight-item MISS-HP scale. We included six questions on religiosity and spirituality to determine whether to use the 8-item or 10-item G-MISS-HP, as two items of the latter depend on religiosity.The questionnaire ended with an open commentary field.

### 2.3. Data Collection

Data were collected using the online survey platform Umfrageonline.com (Enuvo GmBH, Zurich, Switzerland). IP addresses were blinded to the investigators. We used a single survey collector (web address) sent out to participants. Participation was possible using any mobile device (smartphone, tablet, or PC).

#### 2.3.1. Study Size

We aimed to explore at least 100 participants for the pretest study [38,39].

#### 2.3.2. Statistics

We tested the internal consistency of the survey and its sub-questionnaires with the calculation of α. We performed bivariate correlation analysis using Spearman’s rho (*ρ*). Additionally, we conducted multiple linear regression analysis to explore the associations between age, personality traits, G-SVESTR, and G-MISS-GP, with symptom count following the SV experience. Symptom count served as a criterion variable, with each group of predictors added to the model in separate steps. Initially, we added age, followed by the five personality traits (BFI-10), G-SVESTR score, and G-MISS-HP. Multicollinearity diagnostics were performed by inspecting the bivariate correlation matrix as well as tolerance and the variance inflation factor (VIF). In the presence of multicollinearity, predictor variables were mean-centered prior to conducting regression analyses [40]. Correlation analyses and multiple linear regressions were performed with bootstrapping to ensure robust standard errors and confidence intervals (bias-corrected and accelerated (BCa) method based on 1000 bootstrap samples and a confidence interval (CI) of 95%) [41,42]. In addition, bootstrapped coefficient estimates, and CIs are less affected by outliers [43]. Therefore, the results of 95% BCa CIs were considered valid for the hypothesis testing, independent of the reported *p*-value [44,45]. Furthermore, we tested the indirect effects of variables by examining changes in effect size as additional predictors were included in the equation. Indirect effects were tested using Model 4, and interaction effects involving collegial, supervisory, and institutional support as well as physical or psychological distress on symptom count were analyzed using Model 1 of SPSS PROCESS Macro following the approach outlined by Hayes 2018 (version 4) [46]. Depending on sample size, normal distribution, and variance homogeneity, we employed nonparametric and parametric tests for further comparisons. We conducted the quantitative analysis using IBM SPSS v.29.0 (IBM, Armonk, NJ, USA). A *p*-value lower than 0.05 (double-sided) was considered significant.

## 3. Results

### 3.1. Participants

Of 108 participants, 90 completed the SeViD-questionnaire, 72 completed the G-SVESTR, 71 the MISS-HP, and 67 completed all three (52.8%). Completion times ranged from 6 min to 11 h (mean (*M*) = 50 min, standard deviation (*SD*) = 4 h, median = 14 min) with three extreme outliers at 1.3, 1.5, and 11 h.

Of all participants, 63.3% were female. Four percent were not originally trained as GPs but had switched to the field, 7.3% were GPs in training, 55% were GPs, 13.8% were internal medicine physicians working as GPs, and 17.4% were HCAs. Others included GP-associated individuals working in research or administration. In terms of workplace, 45.9% worked in joint GP practices, 40.4% in single practices, and 4.9% in ambulatory healthcare centers. Other workplaces included universities, education, and research, and one GP worked in a hospital. Of all responders, 23.9% had an additional qualification in palliative care. All responders worked in Germany (100%). Objective age ranged from 21 to 68 years (*M* = 47.6 years, *SD* = 11.5).

### 3.2. Descriptive Data

The measurement of personality traits showed normally distributed data for the BFI-10, indicating medium levels of extraversion (*M* = 3.35, *SD* = 0.88), low neuroticism (*M* = 2.75, *SD* = 0.88), high openness (*M* = 3.72, *SD* = 0.94), high conscientiousness (*M* = 4.11, *SD* = 0.78), and medium agreeableness (*M* = 3.43, *SD* = 0.76).

Ninety-two participants reported having experienced one up to 14 symptoms in the case of adverse events, with an average count of *M* = 6 symptoms and *SD* of 3.8.

The instruments demonstrated satisfactory internal consistency, with a high α of 0.91 for the G-SVESTR and an acceptable α of 0.73 for the eight-item G-MISS-HP. Subscales for the G-SVESTR showed to be satisfactory with the exception of collegial support (α = 0.84 for psychological distress, α = 0.82 for physical distress, α = 0.47 for collegial support, α = 0.82 for supervisor support, α = 0.68 for institutional support, α = 0.87 for professional self-efficacy, α = 0.83 for turnover intentions, α = 0.68 for absenteeism, α = 0.68 for absenteeism and α = 0.98 for resilience).

In addition, participants expressed irritability in the free-text entries regarding religious and spiritual questions.

### 3.3. Main Results

Among 82 responders, prevalence of SV-related symptoms was high: Fear of exclusion by colleagues (30.4%), fear of losing their job (12%), listlessness (20.7%), depressive mood (31.5%), concentration difficulties (37.0%), reliving the situation in similar professional situations (22.8%), reliving the situation outside professional activity (32.6%), aggressiveness (4.3%), defensiveness, overly cautious behavior (41.3%), psychosomatic symptoms such as headache and back pain (25.0%), insomnia or excessive need for sleep (44.6%), use of drugs or alcohol (10.9%), feelings of shame (27.2%), feelings of guilt (50.0%), self-doubt (62.0%), social isolation (7.6%), anger towards others (20.7%), anger towards oneself (29.3%), desire for support (40.2%), and desire to process the event for better understanding (46.7%).

Of 91 responders, 16 (17.6%) reported to have experienced symptoms of the phenomenon for more than 12 months. Of 90 responders, 75 (83.3%) reported fully recovering from the event.

The G-MISS-HP score, excluding faith-related questions, used eight instead of ten items ranging from seven to sixty points (not normally distributed, *M* = 25.7, *SD* = 12.0, median = 22). Thirty-three percent of all participants scored above the cut-off value of 28.5 points [32]. Items 9 and 10 were extracted due to no adequate response option for atheists or agnostics to validly answer these items, as suggested in the validation study, and due to free-text entries on the questionnaires. We excluded two items (9 and 10) for all participants and not only for individuals who claimed to be religious. This approach was taken to avoid the challenge of managing two distinct sum scores and the potential reduction of sample power due to the necessity of conducting every analysis (including MI) twice. Furthermore, it is worth noting that several participants expressed irritation with religious and spiritual questions in free-text entries, posing a potential risk of bias.

The G-SVESTR sum score ranged from 11.4 to 32.2 points (normal distribution, *M* = 22.1, *SD* = 5.7). Sub-scores are displayed in Table 1.

Bivariate correlation analysis showed a strong and significant positive correlation between G-SVESTR and G-MISS-HP (*ρ* = 0.65, *p* < 0.001). Additionally, there was a strong positive correlation between G-SVESTR and symptom count (*ρ* = 0.6, *p* < 0.001). Moderate negative correlations were observed between G-SVESTR and objective age (*ρ* = −0.35, *p* = 0.004) and G-SVESTR and conscientiousness (*ρ* = −0.33, *p* = 0.007). Additionally, we found a moderate positive correlation between G-SVESTR and neuroticism (*ρ*= 0.34, *p* = 0.007).

Additional correlations were found for G-MISS-HP, which negatively correlated with extraversion (*ρ* = −24, *p* = 0.04), and positively correlated with neuroticism (*ρ* = 0.32, *p* = 0.01), agreeableness (*ρ* = 0.30, *p* = 0.014), and symptom count (*ρ* = 0.52, *p* < 0.001).

Symptom count was strongly positively correlated to G-MISS-HP and G-SVESTR, as shown above. Additionally symptom count exhibited a week negative correlation with age (*ρ* = −0.25, *p* = 0.04) and a moderate positive correlation with neuroticism (*ρ* = 0.39, *p* < 0.001).

Mann–Whitney U tests showed no significant differences between physicians and HCAs in G-MISS-HP and G-SVESTR sum scores. When comparing survey completers with noncompleters, we detected no difference between these groups for G-MISS-HP and G-SVESTR sum scores and personality traits, except for neuroticism. Using a *t*-test, neuroticism was significantly (*p* = 0.02, Cohen’s *d* = −0.47) lower in the completer group (*M* = 2.65, *SD* = 0.88) than in the noncompleter group (*M* = 3.16, *SD* = 1.3).

Results of univariate regression showed that age was a significant predictor of symptom count after its addition to the equation (unstandardized regression coefficient (*B*) = −0.08, BCa 95% CI [−0.14, −0.03], Table 2). However, after the inclusion of personality traits in the regression model, the impact of age was no longer significant (*B* = −0.04, BCa 95% CI [−0.11, 0.02]). The only significant predictor was neuroticism (*B* = 1.64, BCa 95% CI [0.68, 2.53], Table 3). Therefore, we assumed that age could affect symptom count via neuroticism. After the inclusion of G-SVESTR in the equation, neuroticism (*B* = 1.24, BCa 95% CI [0.29, 2.15]), agreeableness (*B* = 0.95, BCa 95% CI [0.05, 1.78]), and G-SVESTR (*B* = 0.32, BCa 95% CI [0.17, 0.50]) were significant predictors of symptom count (Table 4). Finally, we added the G-MISS-HP score to the model. The significant predictors in the final model were neuroticism (*B* = 1.6, BCa 95% CI [0.78, 2.52]), agreeableness (*B* = 1.62, BCa 95% CI [0.69, 2.44]), G-SVESTR (*B* = 0.23, BCa 95% CI [0.03,0.43], and G-MISS-HP (B = 0.08, BCa 95% CI [0.01, 0.16], Table 5).

As age’s magnitude of effect size diminished after the five personality traits were included in the regression model, and G-SVESTR’s regression coefficient reduced after G-MISS-HP was included in the regression equation, we tested the indirect effect of age on symptom count via five personality traits and the indirect effect of G-SVESTR on symptom count via G-MISS-HP (see Figure 1 and Figure 2).

We performed the first mediation analysis with age as a predictor, the Big Five dimensions as mediators, and symptom count as an outcome.

The results revealed a negative significant indirect effect of age on symptom count *(B* = −0.03, bootstrapped 95% boot CI [−0.06, −0.005]) via neuroticism. Furthermore, the direct effect of age on symptom count in the presence of the mediator neuroticism was found to be nonsignificant (*B* = −0.04, bootstrapped 95% boot CI [−0.11, 0.003]). We observed no further significant indirect effects of age via the other four personality traits on symptom count. The total (indirect + direct) effect of age on symptom count was *B* = −0.08, bootstrapped 95% boot CI [−0.15, −0.01]). In conclusion, neuroticism fully mediated the relationship between age and symptom count. Age and neuroticism account for 23% of the variance of symptom count (*F*(6, 82) = 3.99, *p =* 0.01). Table 6 summarizes the results of the mediational analysis.

A second mediational analysis was performed with G-SVESTR as a predictor, G-MISS-HP as a mediator, and symptom count as an outcome variable. Age and the Big Five personality traits served as covariates.

A significant positive indirect effect was observed for G-SVESTR on symptom count through MI score (B = 0.11, 95% boot CI [0.01, 0.23]). The direct effect of G-SVESTR on symptom count was also significant B = 0.21, 95% CI [0.04, 0.38]. The total effect was B = 0.32, 95% boot CI [0.18, 0.46]. In conclusion, MI partially mediated the association of G-SVESTR with symptom count. The whole model explained 46% of symptom count variance *(F*(7, 61) = 7.54, *p <* 0.001).

In addition, we tested the moderation effect of collegial, supervisory, and institutional support on the associations between physical and psychological distress (subscales of G-SVESTR) and symptom count. The only significant interaction effect was between collegial support and physical distress even after controlling for age and personality traits (Table 7). The model explained 52% of symptom count variance ((*F*(9, 67) = 8.13, *p* < 0.001).

Simple slope analyses revealed that there was no linear relationship between physical distress as a subdimension of G-SVESTR and symptom count when collegial support was low. As opposed to low collegial support, G-SVESTR exhibited a positive correlation with symptom count at high levels of collegial support (see Figure 3). It is interesting to note that, at a low level of physical distress, collegial support can be beneficial for affected individuals, reducing symptom count. However, when physical distress is high, collegial support can no longer buffer the effect of physical distress on symptom count.

## 4. Discussion

### 4.1. Key Results

This is the first study examining the intersectoral transfer of the G-MISS-HP, SeViD, and G-SVESTR questionnaires from the hospital sector to the sector of GPs. There is growing evidence of a psychological burden on HCPs coupled with an anticipated growing shortage of medical resources due to factors such as an aging population, demographic change, climate change, geopolitical instability, inflation, pandemics, and evolving technological complexity. These factors result in heightened demands for physical and psychological support for patients. Consequently, it is crucial to assess SVP and identify the transferability of in-hospital instruments.

We identified that all three surveys are suitable for implementation in the assessment of SVP and MI in GPs and HCAs, although some adjustments seem necessary:

#### 4.1.1. Collegial Support–Optional

GPs and HCAs successfully completed and provided valid responses to the G-SVESTR; however, challenges emerged concerning items associated with “collegial support.” There was a noticeable decrease in internal consistency, and qualitative responses indicated that some participants, particularly those in single practices (comprising 40.4% of all responders), may not have colleagues or supervisors to consult. Consequentially, in future surveys, it is recommended to either make the “colleague” items optional for individuals without direct contact with colleagues or explore inquiries about other peer-support programs within the GP community or support from not work-related peers such as spouses and family members [47].

#### 4.1.2. Secular Version of G-MISS-HP

The eight items of MISS-HP exhibited acceptable properties for GPs and HCAs. However, the 10-item version includes questions for spiritual and religious beliefs, which may result in irritation and impede the survey’s completion. This aligns with reported low religiosity among HCPs in Germany [48] consistent with the findings in the G-MISS-HP validation [32].

#### 4.1.3. “Subjective” and “Objective” SVP

The use of the modified SeViD-questionnaire [30] in our survey also yielded acceptable results. Notably, we did not directly inquire whether respondents identified themselves as an SV. Interestingly, most participants reported experiencing SV symptoms after an event, in contrast to other SeViD surveys [13,14,15,16], where symptom-related items were only accessible if participants identified themselves as SVs. Our findings of a high symptom count may be valid and comparable to that of Austrian pediatricians [16]. Alternatively, the neglect in self-diagnosis could be attributed to an overconfidence effect in SVP [49] or an unintended violation of one’s self-perception. This is comparable to findings in PTSD, where patients may neglect the significance of the disease [50]. This raises the question of miscalibration of “subjective” and “objective” SVP, suggesting the possibility of over- [17] or underassessment [49] of SV. Consequently, there is a need to include both questions (self-diagnosis and an independent assessment of symptoms) in future surveys to achieve better calibration and evaluate neglect in SVP.

#### 4.1.4. G-SVESTR Sum Score

For the first time, we computed an overall G-SVESTR score derived from its subscales. We showed that the usage of the score could offer advantages in multivariate analyses compared with using individual subscales to measure experienced distress caused by an adverse event. Employing the overall score helps mitigate issues of multicollinearity, leading to increased accuracy in measuring SV experiences. Further studies could establish a cut-off score, similar to the MISS-HP instrument, indicating a significant and clinically relevant functional impairment.

#### 4.1.5. The Role of Personality Traits

Our survey revealed that completers did not significantly differ from noncompleters, except for neuroticism in the noncompletion group. The role of personality traits in motivation to participate in online surveys is well documented [51,52].

Building on the third point, it is important to note that neuroticism is identified as a risk factor for experiencing SVP [14]. This raises the possibility of underestimation of the prevalence of individuals who do not complete the survey. Consequently, the inclusion of the BFI-10 in future studies is recommended for a more comprehensive understanding and anticipation of SVP prevalence.

Moreover, the role of personality traits in SV assessments remains significant, as neuroticism emerged as a strong modulator of SV and MI experiences. This finding is in accordance with the findings of SeViD-III and SeViD-A1 studies, whereas neuroticism mediated the association between the length of professional experience and symptom load. Longitudinal studies suggest that neuroticism tends to decrease with age [53]. This implies that younger HCPs might be at higher risk of becoming SVs. Consequently, younger professionals might have a higher demand for prevention and (early) intervention.

#### 4.1.6. The Role of Trauma, Depression, and PTSD

In this survey, we were unable to assess whether participants were SVs or were experiencing other conditions such as PTSD, depression, or other primary psychiatric disorders. During the symptom assessment, over 10% reported substance abuse, which should be further differentiated in future studies. Sleeping disorders, depressive mood (not to be confused with depression), listlessness, and the desire for support showed a demand for further differentiation of second victims (as a normal psychological reaction), second victims that have advanced to “wounded healers” [54] potentially suffering from primary psychiatric conditions, and dysfunctional individuals [7]. Achieving this differentiation could be facilitated by including short questionnaires on depressive symptoms [55,56], balancing the need for additional insights without imposing a high response burden.

#### 4.1.7. Influence of MI and SV on Symptom Count

Our results indicated that MI and SV are related yet distinct constructs, representing different forms of work-related psychological strain. A deeper analysis revealed that symptom count depends on the occurrence of SV and subsequent development of MI. However, MI might not be the cause of SVP; instead, SVP might be a precursor to MI, contributing to the aggravation of symptoms. Considering the phenomenology and case development of SVP [4,7], we postulate that early coping with SVP may play a crucial role in preventing the development of MI. This in turn mitigates the worsening of overall symptoms, encompassing emotions, feelings, and physical and psychological phenomena as well as preventing dysfunctionality and eventual system dropout.

#### 4.1.8. The Role of Collegial Support in Coping with SVP

The identification of the impact of perceived peer support on physical distress score, as revealed in slope analysis, is an important finding in understanding the role of collegial support in SVP. This suggests that when the physical burden stemming from the SV experience is low, high collegial support can significantly decrease symptom count. On the contrary, when individuals encounter substantial physical distress, the impact of collegial support on symptom count seems to be less significant. As a result, peer support becomes particularly vital for those with a lower burden, playing a crucial role in symptom reduction. However, in instances of highly burdened colleagues, peers must be trained to recognize this and initiate the next step of professional support, as detailed in existing literature [7].

Nevertheless, further investigations should explore how environmental factors may either reduce or increase the likelihood of the occurrences of SV or MI, which can eventually lead to additional health impairments and influence coping when an adverse event occurs (see Figure 4). Consequently, a successive questionnaire could be designed with the aim of better distinguishing between symptoms and assessing environmental factors.

## 5. Limitations

In this study, a significant challenge is presented by the small sample size and the utilization of a convenience sampling method [38], potentially leading to selection bias and limiting the generalizability of findings concerning SVP prevalences while not compromising the assessment of the instrument’s properties. However, the study aimed to refine our instrument for a more extensive survey within a larger collective.

Additional limitations such as participants’ irritation due to spiritual questions and the potential influence of personality traits on motivation to participate have been previously noted. Given the cross-sectional design of the study, causal interpretations between the variables are not feasible. However, our conclusions are guided by theoretical considerations and grounded in the findings of previous studies.

## 6. Conclusions

In this study, we demonstrated the transferability of three instruments from the clinical sector to the ambulant sector: G-SVESTR, G-MISS-HP (the 8-item version), and SeViD with a modification to the symptoms question (subjective vs. objective SVP). Additionally, the inclusion of the BFI-10 and potential use of a depression screening tool for the assessment of a depressive disorder should be considered in future research. Concerning the role of MI development and its influence on symptom count, further work including an exploration of symptom course and intensity is essential to comprehend early and late onset symptoms and their progression. Furthermore, it is crucial to assess the role of environmental responses in SVP and MI and their relationship to opportunities and barriers in psychological support strategies (e.g., by family members, and the affected patients and their families involved in the adverse event). In addition, representative longitudinal studies in closed populations, employing random sampling procedures, are required to validate the findings of the current study.

In conclusion, the insights of this study may assist SV researchers in designing and calibrating their instruments for SV assessment, with a specific focus on understanding the role of MI in the effect cluster.

## Figures and Tables

**Figure 1 healthcare-12-00351-f001:**
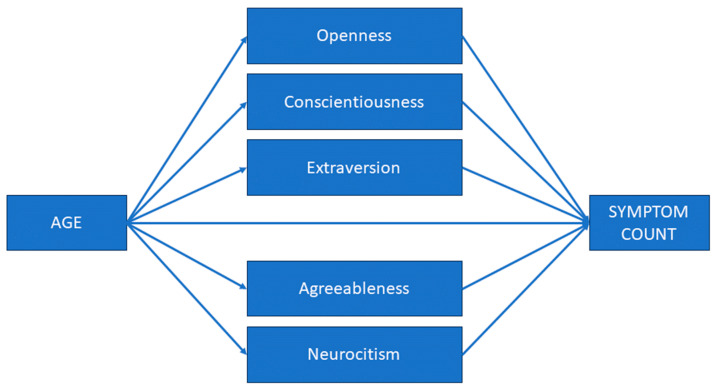
First mediational model. The associations between age and symptom count mediated by the Big Five personality traits (extraversion, agreeableness, openness, conscientiousness, and neuroticism). Symptom count: the sum of symptoms that may result from a second victim experience.

**Figure 2 healthcare-12-00351-f002:**
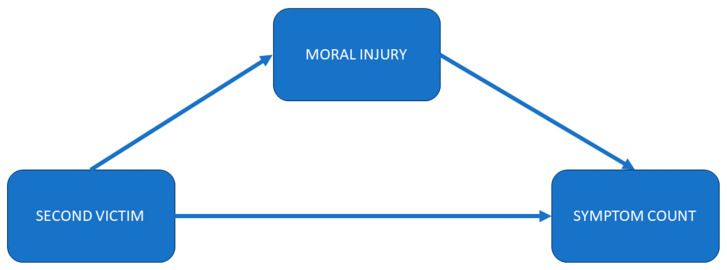
Second mediational model. The associations between second victim experience (G-SVESTR Score) and symptom count mediated by moral injury (G-MISS-HP). G-SVESTR: German version of Second Victim Experience and Support Tool Revised Version, G-MISS-HP: German version of the Moral Injury Symptom and Support Scale for Health Professionals, Symptom count: the sum of symptoms that may result from a second victim experience.

**Figure 3 healthcare-12-00351-f003:**
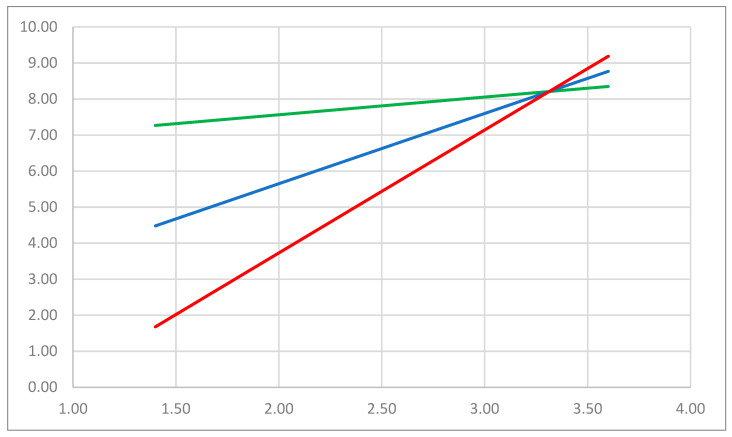
The plot of the simple slope analysis examining the relationship between physical distress (X-axis), a subscale of G-SVESTR as the focal predictor, and symptom count (Y-axis) as the criterion variable across three levels of collegial support, another subscale of G-SVESTR as a moderator. G-SVESTR: German version of the Second Victim Experience and Support Tool Revised Version, Symptom count: sum of the symptoms that can be a further consequence of second victim experience. The red line indicates high collegial support, the blue line moderate collegial support, and the green line signifies low collegial support.

**Figure 4 healthcare-12-00351-f004:**
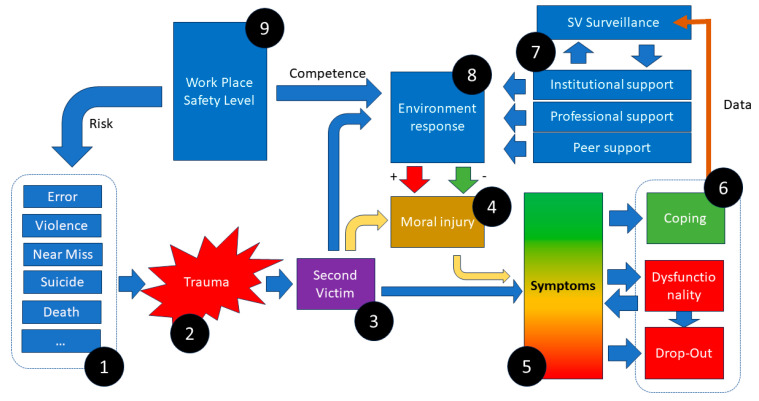
hypothetical framework for the interaction of different elements of SV: Predispositions (1) leading to a traumatic event (2) resulting in SVP (3). To an unknown extent, each environmental reaction (8) and the SVs themselves contribute to MI (4), modulating their emotional reaction and multifaceted early and late symptom count (5), which interacts with outcome (6). Outcomes may stimulate SV surveillance and programs of SV management (7), influencing environmental response (8) and safety levels (9), which in turn affect the risk of reoccurrence of the event.

**Table 1 healthcare-12-00351-t001:** Descriptive statistics for G-MISS-HP and G-SVESTR sum scores, and sub-scores of nine dimensions of G-SVESTR.

Scores	Mean (*M*)	*SD*	Median	Min	Max
MISS-HP-8 item (sum)	24.1	12.2	21	6	57
G-SVESTR (sum)	22.1	5.7	21.3	11.4	32.2
G-SVESTR (Psychological Distress)	3.47	1.06	3.50	1	5
G-SVESTR (Physical Distress)	2.61	1.04	2.60	1	5
G-SVESTR (Colleagial Support)	1.94	0.65	2.00	1	4
G-SVESTR (Supervisor Support)	2.50	1.10	2.5	1	5
G-SVESTR (Institutional Support)	3.14	1.10	3.33	1	5
G-SVESTR (Professional Self-Efficacy)	2.66	1.15	2.75	1	5
G-SVESTR (Turnover Intentions)	1.88	1.03	1.5	1	4.25
G-SVESTR (Absenteeism)	1.82	0.96	1.33	1	4
G-SVESTR (Resilience)	2.10	0.92	2.00	1	4.75

G-MISS-HP: German version of Moral Injury Symptom and Support Scale for Health Professionals, G-SVESTR: German version of the Second Victim Experience and Support Tool Revised Version, *M*: mean, *SD*: standard deviation, Min: minimum value, Max: maximum value.

**Table 2 healthcare-12-00351-t002:** Results of univariate linear regression with age as the predictor and symptom count as the criterion variable.

	Bootstrapping (Bias-Corrected and Accelerated Method Based on 1000 Samples)					
	*B*	Bias	Std. Error	Sig. (2-Tailed)	BCa 95% Confidence Interval	
					Lower	Upper
(Constant)	9.80	0.04	1.65	<0.001	7.03	12.71
Age	−0.08	−0.001	0.03	0.01	−0.14	−0.003

*B*: unstandardized regression coefficient, Std. Error: standard error, BCa: bias-corrected and accelerated bootstrapping method based on 1000 bootstrap samples.

**Table 3 healthcare-12-00351-t003:** Results of multiple linear regression with age and personality traits as predictors and symptom count as the criterion variable.

	Bootstrapping (Bias-Corrected and Accelerated Method Based on 1000 Samples)					
	*B*	Bias	Std. Error	Sig. (2-Tailed)	BCa 95% Confidence Interval	
					Lower	Upper
(Constant)	0.86	−0.04	3.77	0.82	−5.69	8.35
Age	−0.04	0.00	0.03	0.20	−0.11	0.02
Extraversion	−0.01	−0.01	0.44	0.98	−0.88	0.82
Neuroticism	1.64	0.02	0.43	<0.001	0.68	2.53
Openness	0.12	0.00	0.42	0.77	−0.72	0.92
Conscientiousness	−0.15	0.03	0.60	0.80	−1.27	1.18
Agreeableness	0.82	−0.02	0.54	0.13	−0.23	1.84

*B*: unstandardized regression coefficient, Std. Error: standard error, BCa: bias-corrected and accelerated bootstrapping method based on 1000 bootstrap samples.

**Table 4 healthcare-12-00351-t004:** Results of multiple linear regression with age, personality traits, and second victim score as predictors and symptom count as the criterion variable.

	Bootstrapping (Bias-Corrected and Accelerated Method Based on 1000 Samples)					
	*B*	Bias	Std. Error	Sig. (2-Tailed)	BCa 95% Confidence Interval	
					Lower	Upper
(Constant)	−12.12	−0.01	3.91	<0.001	−19.66	−4.34
Age	0.00	0.00	0.03	0.97	−0.07	0.06
Extraversion	0.41	−0.04	0.45	0.37	−0.45	1.19
Neuroticism	1.23	−0.02	0.49	0.01	0.29	2.15
Openness	0.20	0.00	0.39	0.61	−0.61	0.93
Conscientiousness	0.58	0.05	0.54	0.28	−0.46	1.90
Agreeableness	0.95	−0.03	0.49	0.06	0.05	1.78
G-SVESTR	0.32	0.00	0.08	<0.001	0.17	0.50

*B*: unstandardized regression coefficient, Std. Error: standard error, BCa: bias-corrected and accelerated bootstrapping method based on 1000 bootstrap samples, G-SVESTR: German version of Second Victim Experience and Support Tool Revised Version. Please note that we take into consideration the robust BCa 95% confidence interval in significance testing, regardless of the reported *p*-value.

**Table 5 healthcare-12-00351-t005:** Results of multiple linear regression with age, personality traits, second victim, and moral injury score as predictors and symptom count as the criterion variable.

	Bootstrapping (Bias-Corrected and Accelerated Method Based on 1000 Samples)					
	*B*	Bias	Std. Error	Sig. (2-Tailed)	BCa 95% Confidence Interval	
					Lower	Upper
(Constant)	−14.85	0.02	4.21	<0.001	−23.89	−6.31
Age	0.00	0.00	0.03	0.93	−0.07	0.07
Extraversion	0.39	−0.03	0.45	0.38	−0.44	1.14
Neuroticism	1.65	−0.01	0.45	<0.001	0.78	2.52
Openness	0.28	−0.01	0.38	0.44	−0.46	0.94
Conscientiousness	0.34	0.07	0.58	0.55	−0.78	1.77
Agreeableness	1.62	−0.04	0.50	<0.001	0.69	2.44
G-SVESTR	0.23	0.00	0.10	0.03	0.03	0.43
G-MISS-HP	0.08	0.00	0.04	0.03	0.01	0.16

*B*: unstandardized regression coefficient, Std. Error: standard error, BCa: bias-corrected and accelerated bootstrapping method based on 1000 bootstrap samples, G-SVESTR: German version of Second Victim Experience and Support Tool Revised Version, G-MISS-HP: German version of the Moral Injury Symptom and Support Scale for Health Professionals.

**Table 6 healthcare-12-00351-t006:** Unstandardized indirect effects of age on symptom count caused by SV experience via the Big Five personality traits.

	Unstandardized Effect	BootLLCI	BootULCI
Total	−0.04	−0.08	−0.009
Openness	−0.0001	−0.008	0.009
Conscientiousness	−0.0003	−0.008	0.009
Extraversion	−0.0001	−0.009	0.01
Agreeableness	−0.009	−0.03	0.002
Neuroticism	−0.03	−0.06	−0.005

Criterion variable is symptom count. SV: second victim, openness, conscientiousness, extraversion, agreeableness, neuroticism: Big Five personality traits, BootLLCI, BootULCI: lower (LLCI) and upper (ULCI) limits of 95% confidence interval based on 5000 deviation-correction bootstrap samples.

**Table 7 healthcare-12-00351-t007:** Interaction effect between physical distress and collegial support (both subscales of the G-SVESTR) on symptom count.

	Bootstrapping (Bias-Corrected (BC) Method Based on 5000 Samples)				
	*B*	Std. Error	Sig. (2-Tailed)	BC 95% Confidence Interval	
				Lower	Upper
(Constant)	27.55	7.36	<0.01	12.88	42.22
Physical distress	4.82	4.31	<0.001	2.91	6.73
Collegial support	5.21	1.53	0.01	2.10	−8.34
Interaction	−1.55	0.46	0.01	−2.46	−0.64
Age	−0.03	0.03	0.33	−0.09	0.03
Openness	0.05	0.36	0.89	−0.67	0.76
Conscientiousness	0.26	0.37	0.58	−0.66	1.17
Extraversion	0.21	0.37	0.57	−0.53	0.94
Agreeableness	0.43	0.47	0.36	−0.50	1.36
Neuroticism	0.87	0.40	0.03	0.07	1.67

Results of multiple linear regression with symptom count as a criterion variable, physical distress as a focal predictor, collegial support as a moderator and age, openness, conscientiousness, extraversion, agreeableness, and neuroticism as covariates. *B*: unstandardized regression coefficient, Std. Error: standard error, G-SVESTR: German version of the Second Victim Experience and Support Tool Revised Version, Interaction: product term of physical distress and collegial support, Symptom count: sum of the symptoms that may result from the second victim experience. Lower, Upper: limits of the 95% confidence interval based on 5000 bias-corrected (BC) bootstrap samples. *Note*. The collegial support subscale is not inverted; higher scores are associated with poorer collegial support.

## Data Availability

Data is available upon request.

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
