# Peer review of "Applicability and Validity of Second Victim Assessment Instruments among General Practitioners and Healthcare Assistants (SEVID-IX Study)"

_healthcare, 2024, doi:10.3390/healthcare12030351_

Round 1

Reviewer 1 Report

Comments and Suggestions for Authors

It's an interesting topic, and I would like to make suggestions for fine-tuning of this paper.  Greater clarity and structure are needed - if the authors could more clearly outline the rationale, definitions for key terms they are describing throughout the paper, the reason for selecting the variables they are selecting (for assessment), that would add to the flow of the paper.

Some specific suggestions below:

In the first instance when you mention second victim phenomenon, could you please describe what it is, and please include references for that.

Lines 55-56 - When you are mentioning a prevalence between 40%-89%, could you please clarify what the prevalence is referring to?  For example, are you referring to the prevalence of posttraumatic stress disorder, or is this the proportion of people who are second victims or perhaps something else?  Greater clarity would be needed in this instance.

When you first mention moral injury, could you please describe what it is.

Within the first section of your manuscript - the Background - could you please include more description on the instruments that you are looking to assess. 

In the second part of the Objectives, you mention, "Our objectives were to determinate correlations between demographic characteristics, intensity of experienced moral injury and second victim phenomenon and symptom count. ... etc"  Could you please clarify this a bit more.  Why is it important to look at this?

How did you select the variables for analysis?  Could you please provide a bit more of an explanation for this (with referencing)?

Could you please state the validity and reliability of the instruments that you are focusing on?

Good luck with your future work!

Comments on the Quality of English Language

Editing of the paper is needed.

Author Response

Reviewer #1

It's an interesting topic, and I would like to make suggestions for fine-tuning of this paper.  Greater clarity and structure are needed - if the authors could more clearly outline the rationale, definitions for key terms they are describing throughout the paper, the reason for selecting the variables they are selecting (for assessment), that would add to the flow of the paper. Some specific suggestions below:

Thank you very much for this feedback. We revised the manuscript according to the following suggestions to add more clarity.

In the first instance when you mention second victim phenomenon, could you please describe what it is, and please include references for that.

We added the definition by the European Research Network on Second Victim (ERNST) in the introduction section and the references to the definition.

Lines 55-56 - When you are mentioning a prevalence between 40%-89%, could you please clarify what the prevalence is referring to?  For example, are you referring to the prevalence of posttraumatic stress disorder, or is this the proportion of people who are second victims or perhaps something else?  Greater clarity would be needed in this instance.

We added the precise citations and percentages of these studies that refer to the second victim effect, but not to PTSD. We lined this out more clearly.  

When you first mention moral injury, could you please describe what it is.

We added references and explanations on moral injury to the introduction section.

Within the first section of your manuscript - the Background - could you please include more description on the instruments that you are looking to assess. 

We added a deeper explanation of the instruments used and added references to this.

In the second part of the Objectives, you mention, "Our objectives were to determinate correlations between demographic characteristics, intensity of experienced moral injury and second victim phenomenon and symptom count. ... etc"  Could you please clarify this a bit more.  Why is it important to look at this?

We added some explanations on this issue, mainly to elaborate new hypotheses on the phenomenon with the possible need for different prevention and interventions. 

How did you select the variables for analysis?  Could you please provide a bit more of an explanation for this (with referencing)?

We used the pre-validated surveys with some modifications to the SeVid questionnaires. We described this in more detail in the variables section.

Could you please state the validity and reliability of the instruments that you are focusing on?

We included the references for validity and reliability tests in the English versions and the German translations in the methods sections.

Good luck with your future work!

We thank reviewer #1 for her/his valuable contribution to our work. Thank you very much!

Reviewer 2 Report

Comments and Suggestions for Authors

High importance of the study focused on primary care, a group and health care setting where the issue of the second victim has been little studied. And where it is shown that the three instruments used are transferable from the clinical sector; the Big Five Inventory (BFI-10), the G-SVESTR and the G-MISS-HP.

More scientifically interesting is the attempt to analyse the transferability of the scales to the collective, with positive results.

Our aim was to

to clarify the transferability and to identify useful and essential modifications of the surveys with regard to structural properties or content validity.

to determine correlations between demographic characteristics, the intensity of moral harm experienced and the phenomenon of second victim and symptom count.

to test the impact of age, personality traits, second victim score and moral injury on symptom count and to determine the possible protective role of collegial, supervisory and institutional support on the associations between the psychological and physical distress subscales of the G-SVESTR and symptom count. 

To test the impact of age, personality traits, second victim score and moral injury on symptom counts and to determine the possible protective role of collegial, supervisory and institutional support on the associations between the psychological and physical distress subscales of the G-SVESTR and symptom counts. 

All three have been tested: according to the results they show that the group experiences the same problems as other health professionals in relation to recall symptoms in adverse situations, and in all subscales of the G-MISS-HP . Also the results on the prevalence of symptoms related to the second victim were high and the crosstabulations with sociodemographic variables and the BFI-10 show how age could affect the symptom count through neuroticism.

Insufficient support by institutional (colleagues of equal or higher rank, derived from the structure of care (consultations where there is only one professional).

Being aware that the aim was to analyse the validity of the instrument and as they highlight the recommendation to revise the term colleagues in the face of this group of professionals who usually work in isolation, they account for a significant 40.4%, as indicated in row 327, also evidence ( which the need to contemplate these situations, to remedy this lack of institutional support.

The study, in general, demonstrates the need for psychological support and reparation of moral damage to primary care professionals and the lack of resources for this purpose.

The method is concerned about the online response and recruitment through professional networks which is not sufficiently explained (how the contact was made and the selection of the candidate), whether or not there is understanding of the items, but given the amount of sample and the difficulties of access to it, the use of this procedure is understood.

They are recruited through professional networks and

The questionnaire does not foresee this, but it does not foresee that the second victim is a victim of a crime and that the second victim is a victim of a crime,

The questionnaire does not foresee this, the fact that there are other contextual forms that the authors do include in the study, important adaptation to the context of the reality of family doctors.

The study assumes that 417 encounters is too small a sample to propose categorically the suitability of the instruments, but it is true that in the sample used, the majority of family doctors have similar results, which opens up the need to continue exploring in the same direction as the study suggests.

It is recommended to clarify the procedure and some concepts (second victim and impact on professionals) in the introduction or theoretical framework in order to facilitate reading and understanding.

For this purpose, they recommend a number of example studies

Coughlan, B. Powell, D., and Higgins, M.F. (2017) The Second Victim: a Review, European Journal of Obstetrics & Gynecology and Reproductive Biology, Volume 213, Pp 11-16. https://doi.org/10.1016/j.ejogrb.2017.04.002

Rebecca R. Quillivan, Jonathan D. Burlison, Emily K. Browne, Susan D. Scott, James M. Hoffman, (2016). Patient Safety Culture and the Second Victim Phenomenon: Connecting Culture to Staff Distress in Nurses. The Joint Commission Journal on Quality and Patient Safety, Volume 42, Issue 8, Pp 377-AP2, https://doi.org/10.1016/S1553-7250(16)42053-2

Author Response

Reviewer #2

High importance of the study focused on primary care, a group and health care setting where the issue of the second victim has been little studied. And where it is shown that the three instruments used are transferable from the clinical sector; the Big Five Inventory (BFI-10), the G-SVESTR and the G-MISS-HP. More scientifically interesting is the attempt to analyse the transferability of the scales to the collective, with positive results. Our aim was

to to clarify the transferability and to identify useful and essential modifications of the surveys with regard to structural properties or content validity.

to determine correlations between demographic characteristics, the intensity of moral harm experienced and the phenomenon of second victim and symptom count.

to test the impact of age, personality traits, second victim score and moral injury on symptom count and to determine the possible protective role of collegial, supervisory and institutional support on the associations between the psychological and physical distress subscales of the G-SVESTR and symptom count. 

To test the impact of age, personality traits, second victim score and moral injury on symptom counts and to determine the possible protective role of collegial, supervisory and institutional support on the associations between the psychological and physical distress subscales of the G-SVESTR and symptom counts. 

All three have been tested: according to the results they show that the group experiences the same problems as other health professionals in relation to recall symptoms in adverse situations, and in all subscales of the G-MISS-HP . Also the results on the prevalence of symptoms related to the second victim were high and the crosstabulations with sociodemographic variables and the BFI-10 show how age could affect the symptom count through neuroticism.

Insufficient support by institutional (colleagues of equal or higher rank, derived from the structure of care (consultations where there is only one professional).

Being aware that the aim was to analyse the validity of the instrument and as they highlight the recommendation to revise the term colleagues in the face of this group of professionals who usually work in isolation, they account for a significant 40.4%, as indicated in row 327, also evidence ( which the need to contemplate these situations, to remedy this lack of institutional support.

 The study, in general, demonstrates the need for psychological support and reparation of moral damage to primary care professionals and the lack of resources for this purpose.

The method is concerned about the online response and recruitment through professional networks which is not sufficiently explained (how the contact was made and the selection of the candidate), whether or not there is understanding of the items, but given the amount of sample and the difficulties of access to it, the use of this procedure is understood.

They are recruited through professional networks and The questionnaire does not foresee this, but it does not foresee that the second victim is a victim of a crime and that the second victim is a victim of a crime, The questionnaire does not foresee this, the fact that there are other contextual forms that the authors do include in the study, important adaptation to the context of the reality of family doctors.

We thank reviewer #2 for her/his comment on that. Our questionnaire and instrument focus not on crime but on healthcare. According to the definitions, a second victim may be a victim due to violence or even crime – but the definition is not limited to this. We added more detail in the introduction to clarify the definition better.   

The study assumes that 417 encounters is too small a sample to propose categorically the suitability of the instruments, but it is true that in the sample used, the majority of family doctors have similar results, which opens up the need to continue exploring in the same direction as the study suggests.

It is recommended to clarify the procedure and some concepts (second victim and impact on professionals) in the introduction or theoretical framework in order to facilitate reading and understanding.

For this purpose, they recommend a number of example studies

Coughlan, B. Powell, D., and Higgins, M.F. (2017) The Second Victim: a Review, European Journal of Obstetrics & Gynecology and Reproductive Biology, Volume 213, Pp 11-16. https://doi.org/10.1016/j.ejogrb.2017.04.002

 Rebecca R. Quillivan, Jonathan D. Burlison, Emily K. Browne, Susan D. Scott, James M. Hoffman, (2016). Patient Safety Culture and the Second Victim Phenomenon: Connecting Culture to Staff Distress in Nurses. The Joint Commission Journal on Quality and Patient Safety, Volume 42, Issue 8, Pp 377-AP2, https://doi.org/10.1016/S1553-7250(16)42053-2

Thank you very much. We added more detail on the concepts of second victims and the definitions of moral injury and other concepts of psychological burden and added both references to the manuscript. .

Thank you very much.  

Round 2

Reviewer 1 Report

Comments and Suggestions for Authors

Thank you for your previous response.  A few final comments for fine-tuning of this paper, please see below:

When you are mentioning these studies, as in the following statement – please provide references: “Neuroticism was identified as a risk factor for developing SVP by the Second Victims in Deutschland (SeViD)-studies.”

Please state the validity and reliability of the GSVEST-R and G-MISS-HP, with references to published papers provided.

With respect to the following phrase: “This study aims to investigate the usability of the G-SVESTR” – please clarify what you mean by this, and how are you aiming to investigate the usability.

Please state more clearly why it is important to investigate the transferability and usability of these instruments.

In the section called Objectives, it is stated: “This aimed to pave the way for developing new hypotheses concerning the phenomenon…” - please state which phenomenon you are referring to.

There is a segment in the paper, which reads: “we aimed to test the impact of age, personality traits, SV, and MI on the symptom count…” – could you please clarify why you felt it would be important to test for the impact of these variables on symptom count? 

There is a segment in the paper, which reads: “Participants were recruited through the investigators’ private networks” – please provide more information on the process and participants, demographic information would be good too.

Please state in the Discussion section that a limitation of this study is selection bias, due to the way the participants were recruited for this study.  Please also state limited generalizability and that future research should repeat this study using random sampling procedures.

Comments on the Quality of English Language

Moderate editing of English language required

Author Response

Reviewer #1

Thank you for your previous response.  A few final comments for fine-tuning of this paper, please see below:

We appreciate your contribution to improving the quality of the paper. We made the necessary adjustments to refine the manuscript based on your feedback.

When you are mentioning these studies, as in the following statement – please provide references: “Neuroticism was identified as a risk factor for developing SVP by the Second Victims in Deutschland (SeViD)-studies.”

We added the references as suggested.

Please state the validity and reliability of the GSVEST-R and G-MISS-HP, with references to published papers provided.

We provided this information in the revised manuscript. 

With respect to the following phrase: “This study aims to investigate the usability of the G-SVESTR” – please clarify what you mean by this, and how are you aiming to investigate the usability.

To achieve this, we have incorporated an open-field entry method allowing participants to freely express their views on individual items and report any problems with the questionnaire. We also inspected the quantitative data obtained in the study and compared it to previously conducted SeViD studies in the inpatient sector. We report on this in the revised introduction section.

Please state more clearly why it is important to investigate the transferability and usability of these instruments.

We briefly explained the importance of usability investigation in the revised introduction section.

In the section called Objectives, it is stated: “This aimed to pave the way for developing new hypotheses concerning the phenomenon…” - please state which phenomenon you are referring to.

The revised sentence now reads:

“This aimed to pave the way for developing new hypotheses concerning the SVP and MI and their manifestations in disparate situations, professions, and occupational settings – potentially prompting the need for different prevention and education frameworks.” 

There is a segment in the paper, which reads: “we aimed to test the impact of age, personality traits, SV, and MI on the symptom count…” – could you please clarify why you felt it would be important to test for the impact of these variables on symptom count

We briefly explained the reasons for conducting these analyses in the introduction section of the revised manuscript.

There is a segment in the paper, which reads: “Participants were recruited through the investigators’ private networks” – please provide more information on the process and participants, demographic information would be good too.

Thank you for the opportunity to specify the recruitment process, we added the information in the methods section. However, we have no information on demographic characteristics of the individuals who were addressed through our recruitment approach.

Please state in the Discussion section that a limitation of this study is selection bias, due to the way the participants were recruited for this study.  Please also state limited generalizability and that future research should repeat this study using random sampling procedures.

We acknowledged the limitation related to selection bias in the limitation section of the revised manuscript. In the conclusion section, we appeal to researchers to conduct representative studies in closed populations.

Reviewer 2 Report

Comments and Suggestions for Authors

The revision significantly improves the work
The theoretical framework is now perfectly focused, clarifying the concepts it measures and reviews the work, as well as sample search instructions and procedures.
I believe that this is an excellent work, little addressed and that it suggests a much needed line of research in the field of professional health care.

Author Response

The revision significantly improves the work

The theoretical framework is now perfectly focused, clarifying the concepts it measures and reviews the work, as well as sample search instructions and procedures.

I believe that this is an excellent work, little addressed and that it suggests a much needed line of research in the field of professional health care.

Thank you so much for your reflective and encouraging feedback. We truly appreciate your positive comments on the revision.